# Recognition of Vehicle License Plates Based on Image Processing

**Tae-Gu Kim** [1],[†] , **Byoung-Ju Yun** [1],[†] , **Tae-Hun Kim** [2], **Jae-Young Lee** [1], **Kil-Houm Park** [1], **Yoosoo Jeong** [3],[*] and **Hyun Deok Kim** [1],[*]

1   School of Electronics Engineering, IT College, Kyungpook National University, 80, Daehak-ro, Buk-gu, Daegu 41566, Korea; play1472@knu.ac.kr (T.-G.K.); bjisyun@ee.knu.ac.kr (B.-J.Y.); dipvision.ljy@gmail.com (J.-Y.L.); khpark@ee.knu.ac.kr (K.-H.P.)
2   DIPVISION, 80, Daehak-ro, Buk-gu, Daegu 41566, Korea; dipvision.ceo@gmail.com
3   Daegu-Gyeongbuk Medical Innovation Foundation, 88, Dongnae-ro, Dong-gu, Daegu 41061, Korea
*   Correspondence: sysjeong@dgmif.re.kr (Y.J.); hdkim@knu.ac.kr (H.D.K.); Tel.: +82-53-950-7578 (H.D.K.)
†   These authors contributed equally to this work.

**Abstract:** In this study, we have proposed an algorithm that solves the problems which occur during the recognition of a vehicle license plate through closed-circuit television (CCTV) by using a deep learning model trained with a general database. The deep learning model which is commonly used suffers with a disadvantage of low recognition rate in the tilted and low-resolution images, as it is trained with images acquired from the front of the license plate. Furthermore, the vehicle images acquired by using CCTV have issues such as limitation of resolution and perspective distortion. Such factors make it difficult to apply the commonly used deep learning model. To improve the recognition rate, an algorithm which is a combination of the super-resolution generative adversarial network (SRGAN) model, and the perspective distortion correction algorithm is proposed in this paper. The accuracy of the proposed algorithm was verified with a character recognition algorithm YOLO v2, and the recognition rate of the vehicle license plate image was improved 8.8% from the original images.

**Keywords:** deep learning; license plate detection; image processing; SRGAN; CCTV image

## 1. Introduction

Artificial intelligence (AI) technology is a branch of computer science that includes machine learning (ML) and deep learning (DL). AI can be perceived as using any device to imitate the human cognitive processes such as learning, applying, and solving complex problems, etc. [1]. The AI techniques are well suited to imaging-based fields, as an image is the main source of data for training the AI algorithms. ML is dedicated to deploying algorithms. ML algorithms could be viewed as mapping the observed input data variables into the output results [2]. DL is an artificial network system that simulates the concept of human neurons. These techniques have achieved impressive progress in many areas [1,2]. With the development of DL, research on various object recognition methods is being actively conducted, and the vehicle license plate recognition field is one such method.

Vehicle license plate recognition is frequently used for parking management and speed limit enforcement systems. This technology prevents car-related accidents and crimes. The closed-circuit television (CCTV) cameras installed on roads or buildings are used to store and transmit videos and images. According to the statistics from the National Statistical Office of South Korea, about 1.2 million CCTVs are installed in South Korea as of 2019. However, the data acquired through this system is not being fully utilized. It requires much time and human resources to properly utilize and analyze the images captured by CCTV, which becomes a limiting factor in quick response to an accident or crime. This problem can be effectively addressed by deploying an automated method that

can quickly detect information, such as the type, color, and license plate of the vehicle. The technology for recognizing a vehicle license plate uses an image obtained through CCTV, which is usually taken from a long distance and with a wide angle of view. This causes problems, such as a limitation of the resolution, motion blur, and perspective distortion due to various environmental changes and camera installation locations. It is desired to have an image processing technology that can accurately recognizes the characters smaller than the image size. With the development of image processing technology, studies have been conducted in areas such as object detection, tracking, and camera captured image recognition technologies [3]. For efficiently managing the vehicles information, research on vehicle license plate recognition is also being actively conducted [4,5].

In general, the public license plate recognition model is trained based on the front license plate image. The model trained using this technique suffers with the issues of decreased recognition rate when the license plate is recognized in the vehicle image captured using CCTV [6,7]. This is due to the fact that when a vehicle image is captured in a general environment using CCTV, the license plate is tilted, or recorded in low resolution. A common way to solve this problem is to develop a new database and retrain it to solve the recognition rate problem. However, this requires a high cost and time. To overcome this problem, we have proposed an image processing algorithm which can improve the license plate recognition rate by using the existing database.

In CCTV images, the license plates usually appear to be tilted and have low-resolution that results in the lower recognition rate when the existing learning model is applied. To solve this problem, this paper proposed the resolution improvement method and the method for restoring a tilted image. The proposed technique consists of two steps that enhances the recognition rate of the license plate. In the first step, the deep learning-based super-resolution generative adversarial network (SRGAN) algorithm [8] is used to improve the lower resolution image of the license plate. In the second step, the perspective transformation technique is deployed on the tilted license plate image for the correction of perspective distortion.

Section 2 of this paper, the 'License Plate Recognition System', contains the proposed license plate recognition system which is composed of the SRGAN method and perspective distortion. Section 3 is the 'Experimental Results', which provides the experimental verification of the proposed algorithm, followed by Section 4, our 'Conclusion'.

## 2. License Plate Recognition System

### 2.1. System Configuration

In this paper, we propose a novel method to improve the recognition rate of the deep learning models trained with a general database for vehicle license plate recognition using CCTV images. Figure 1 shows the flowchart of the proposed license plate recognition system. First, a commonly used deep learning model YOLO v2 [9] is applied to the input image. This model is used to acquire an image, including a license plate area, which is the area of interest [9]. From the detected ROI image, a high-resolution license plate image is obtained using SRGAN. The license plate area is extracted, and the boundary pixels of the license plate are obtained by using the image segmentation technique. A linear approximation method is applied to the boundary pixels, and a separate linear equation is used for each side of the license plate. The intersection points of all straight lines are calculated by using the above-applied set of linear equations. The homography between the feature points is calculated, and the perspective distortion is corrected. Finally, the YOLO v2 based character recognition technique is applied to the corrected image for recognizing the license plate character.

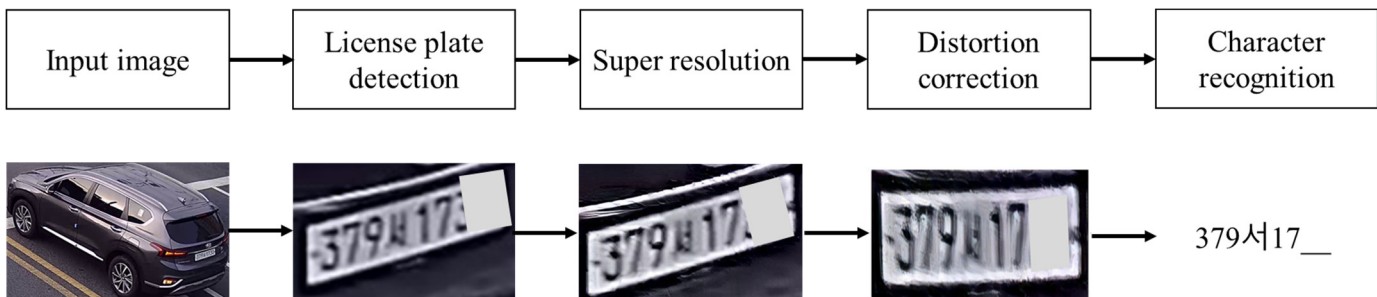

**Figure 1.** Flowchart for the license plate recognition algorithm.

### 2.2. License Plate Detection

As there is much unnecessary information, such as noise and background area, in the entire vehicle image, it is not easy to recognize the license plate. To solve this problem, the algorithm must detect the area of the vehicle license plate, which is the area of interest. In this study, we used a pre-trained YOLO v2 network model for vehicle license plate detection. Table 1 shows the YOLO v2 architecture. Figure 2 shows the result of detecting the license plates using pre-trained YOLO v2 [9,10].

**Table 1.** YOLO v2 architecture.

| Type | Filters | Size/Stride | Output |
|---|---|---|---|
| Convolutional | 32 | $3 \times 3$ | $224 \times 224$ |
| Maxpool | | $2 \times 2/2$ | $112 \times 112$ |
| Convolutional | 64 | $3 \times 3$ | $112 \times 112$ |
| Maxpool | | $2 \times 2/2$ | $56 \times 56$ |
| Convolutional | 128 | $3 \times 3$ | $56 \times 56$ |
| Convolutional | 64 | $1 \times 1$ | $56 \times 56$ |
| Convolutional | 128 | $3 \times 3$ | $56 \times 56$ |
| Maxpool | | $2 \times 2/2$ | $28 \times 28$ |
| Convolutional | 256 | $3 \times 3$ | $28 \times 28$ |
| Convolutional | 128 | $1 \times 1$ | $28 \times 28$ |
| Convolutional | 256 | $3 \times 3$ | $28 \times 28$ |
| Maxpool | | $2 \times 2/2$ | $14 \times 14$ |
| Convolutional | 512 | $3 \times 3$ | $14 \times 14$ |
| Convolutional | 256 | $1 \times 1$ | $14 \times 14$ |
| Convolutional | 512 | $3 \times 3$ | $14 \times 14$ |
| Convolutional | 256 | $1 \times 1$ | $14 \times 14$ |
| Convolutional | 512 | $3 \times 3$ | $14 \times 14$ |
| Maxpool | | $2 \times 2/2$ | $7 \times 7$ |
| Convolutional | 1024 | $3 \times 3$ | $7 \times 7$ |
| Convolutional | 512 | $1 \times 1$ | $7 \times 7$ |
| Convolutional | 1024 | $3 \times 3$ | $7 \times 7$ |
| Convolutional | 512 | $1 \times 1$ | $7 \times 7$ |
| Convolutional | 1024 | $3 \times 3$ | $7 \times 7$ |
| Convolutional | | $1 \times 1$ | $7 \times 7$ |
| Avgpool | 1000 | Global | 1000 |
| Softmax | | | |

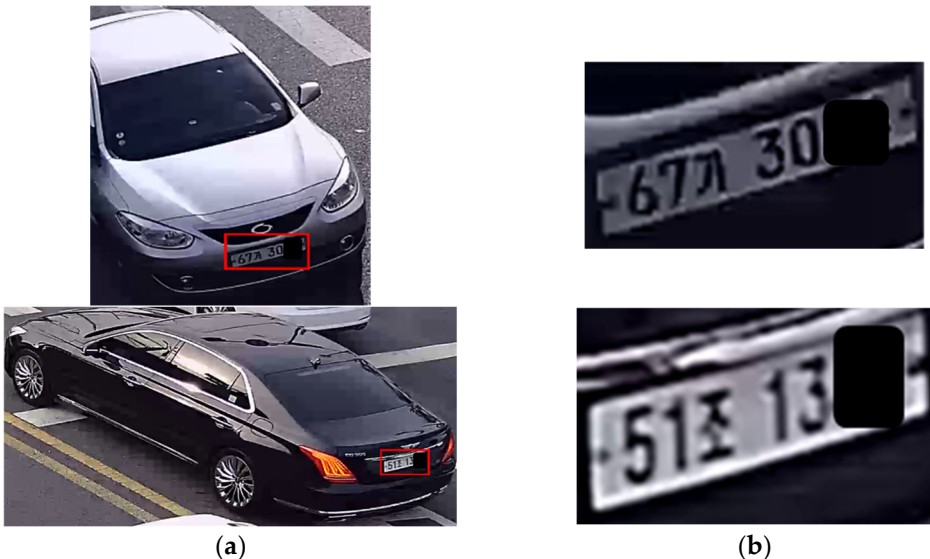

**Figure 2.** (**a**) Input images; (**b**) license plate detection images.

*2.3. Super Resolution*

2.3.1. Super-Resolution Generative Adversarial Network (SRGAN)

A generative adversarial network (GAN) is a deep learning model consisting of a generator and discriminant network. The delimiter learns to separate the actual data from the generated data, while the constructor learns in the direction that interferes with it. In this process, the generator expects to discover the manifold where the actual sample distribution exists. The unsupervised learning GAN has the characteristic of estimating the probability distribution of the original data, and allowing the artificial neural network to create the distribution. Figure 3 shows the network structure of SRGAN using two deep learning models with opposing relationships. This algorithm is used in paper [8].

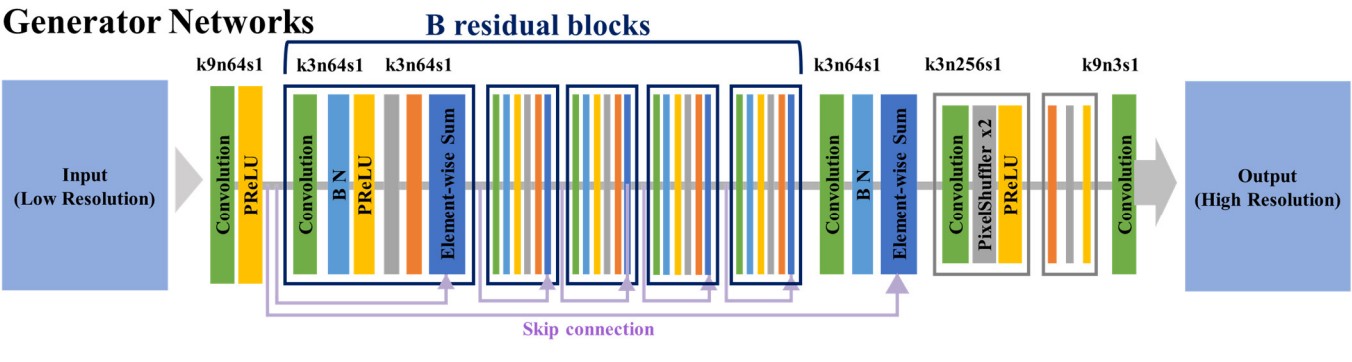

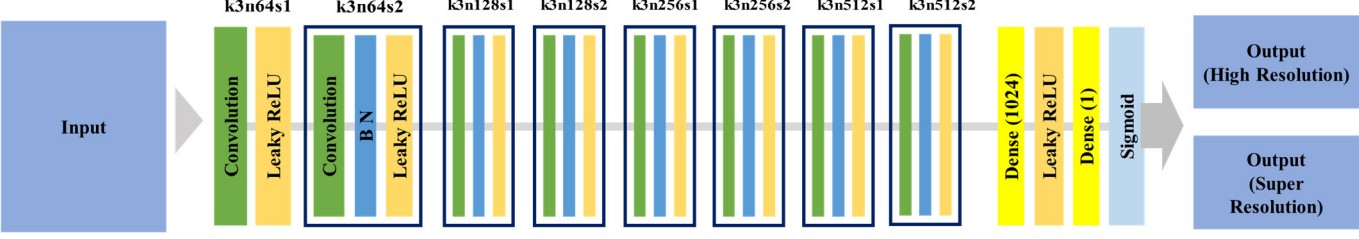

**Figure 3.** SRGAN network architecture.

The generator's deep learning model receives and trains with low-resolution images. The generator inference results are fed into the discriminator and learned until the high-

resolution image is determined. The learned generator makes it possible to reconstruct a low-resolution image to deceive the discriminator that determines whether it is a high-resolution image. Thus, the generator can perform a high-resolution restoration [11,12].

### 2.3.2. Improved Low-Resolution License Plate Image

In this study, the image was improved using the SRGAN model to recognize a vehicle license plate captured with a low-resolution CCTV. Figure 4 shows the image of the improved license plate using the SRGAN model.

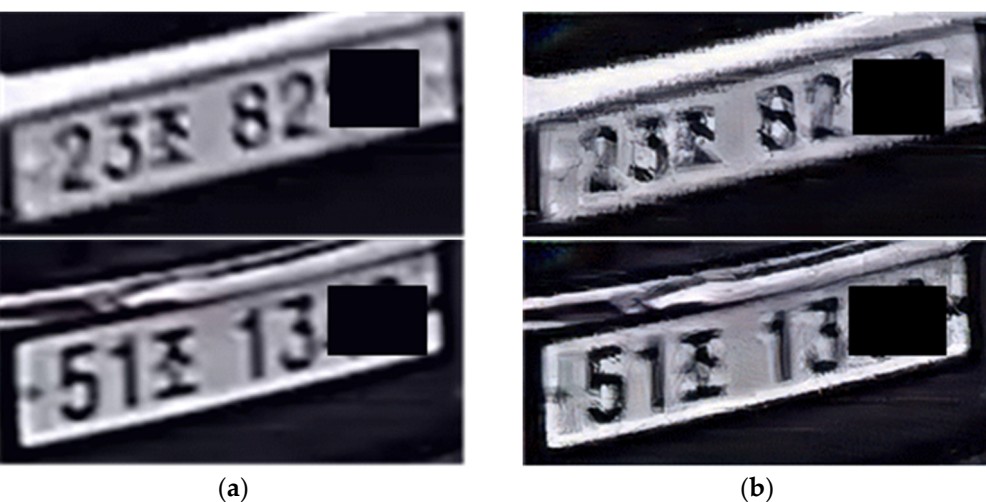

|  (a) | (b) |

**Figure 4.** (**a**) Low-resolution image; (**b**) Improved image.

It can be seen in Figure 4 that the improved license plate image using SRGAN has sharper edged components, corresponding to the high frequencies than the input image. The improved image is used as the input image for license plate recognition [12].

### *2.4. Distortion Conrrection*

The improved image using SRGAN was used as the input image for the image processing step. This step is used to correct the perspective distortion, which is the problem in the license plate recognition.

### 2.4.1. Binarization

First, the license plate area is binarized for the perspective distortion correction. In the image segmentation Equation (1), $f$ and $g$ are the input and output images, respectively, and $T$ is the threshold value. For the input image $f$, when the value of the pixel $(x, y)$ is less than the threshold value $T$, the corresponding pixel value is 0. However, when the value is greater than or equal to the threshold value, the corresponding pixel values become 1. Thus, the divided image can be obtained.

$$g(\text{x, y}) = \begin{cases} if\ f(x,y)\ \geq T,\ 1 \\ if\ f(x,y)\ < T,\ 0 \end{cases} \tag{1}$$

Figure 5 shows the binarization image. This image is used to detect the parallelogram corresponding to the license plate area. This is an essential technique for selecting a threshold value. It distinguishes the area corresponding to the license plate in the obtained image. In this study, $T$ is selected using the Otsu [13] binarization technique to automatically determine the threshold value.

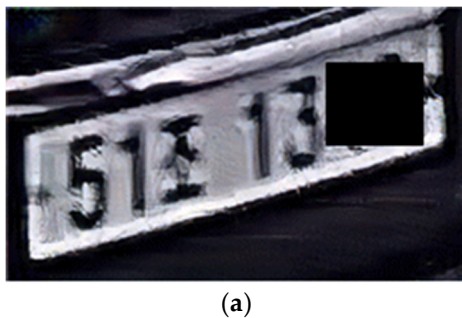 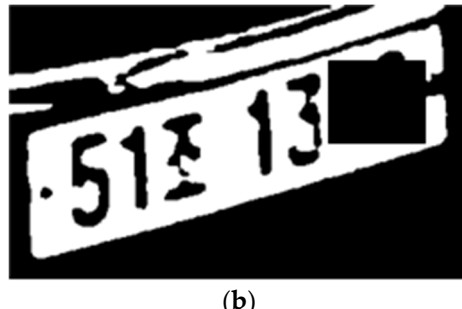

(**a**)                     (**b**)

**Figure 5.** (**a**) Input image; (**b**) Otsu binary image.

### 2.4.2. Morphology

The morphological filter was used to remove noise included in the boundary line of the license plate area, and smooth the boundary line in the binary image. First, the dilation operation was used with a $3 \times 3$ mask as Equation (2) for the Otsu threshold image result $g$. Next, the erosion operation was conducted to remove noise with a size smaller than $6 \times 6$ using a $6 \times 6$ mask, as given in Equation (2). Then, the boundary line in the resulting image of the morphological operation appears close to the straight line, as shown in Figure 6 [14].

$$
\begin{aligned}
I \oplus H &\equiv \{(p+q)|\ for\ every\ p \in I,\ q \in H\} \\
I \ominus H &\equiv \left\{p \in \mathbf{Z}^2 \middle|\ (p+q) \in I,\ for\ every\ q \in H\right\}
\end{aligned}
\tag{2}
$$

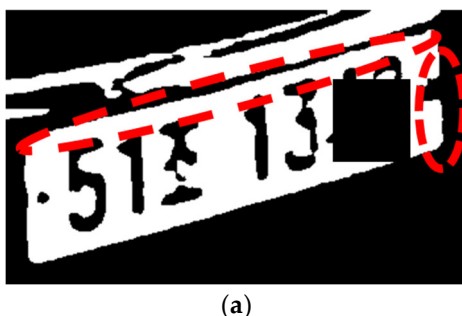 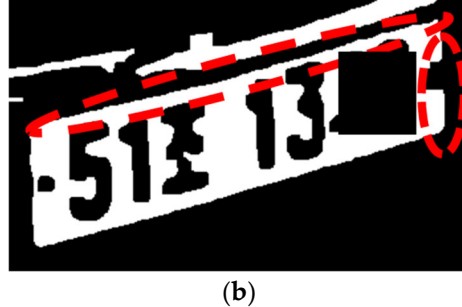

(**a**)                     (**b**)

**Figure 6.** (**a**) Input image; (**b**) Result image after morphology calculation.

### 2.4.3. Random Sample Consensus (RANSAC)

After the morphological operation, the linear approximation is used to detect the quadrangle corresponding to the license plate area. Finding an appropriate linear model for the pixels $(x_i, y_i)$ of the extracted image means the process of minimizing $a$, $b$ in the following equation.

$$
E = \sum_{k=1}^{N} (y_i - (ax_i + b))^2
\tag{3}
$$

The least-square solution can be obtained by calculating the pseudo-inverse matrix, as shown below.

$$
[a\ b] = [y_1 \cdots y_i \cdots y_N]
\begin{bmatrix} x_1 & 1 \\ \vdots & \vdots \\ x_N & 1 \end{bmatrix}
\left(
\begin{bmatrix} x_1 & \cdots & x_i & \cdots x_N \\ 1 & \cdots & 1 & \cdots 1 \end{bmatrix}
\begin{bmatrix} x_1 & 1 \\ \vdots & \vdots \\ x_N & 1 \end{bmatrix}
\right)
\tag{4}
$$

In the above calculated model, the results will depend on the quality of the input data. Thus, we apply the RANSAC algorithm to obtain the optimal solution without including bad data into the model through random sampling [15].

First, we randomly select five points, the minimum data for determining the elliptic model parameters. For five points, the model is obtained using the least-squares method stated earlier. For all data in the image, the number of data is calculated, where the distance between the ellipse and data is less than a predetermined allowance. The data contained within the tolerance is called an inlier, while the data outside the boundary is called an outlier. When the proportion of the inlier in the total data exceeds a certain threshold, the model parameters are determined again by only the inlier. If the percentage of inlier is below a certain level, the previous process is repeated by randomly selecting data.

Each linear approximation technique is an application of the RANSAC line-fitting technique. This is a technique for selecting a reliable pixel with an approximate equation. It is approximated without being affected by other linear components that can be approximated. RANSAC line fitting is applied to the outermost coordinates adjacent to the line $(x_i, y_i)$. The second approximate line equation is obtained in this process. By repeating this process (Figure 7b), four approximate linear equations and four feature points are obtained.

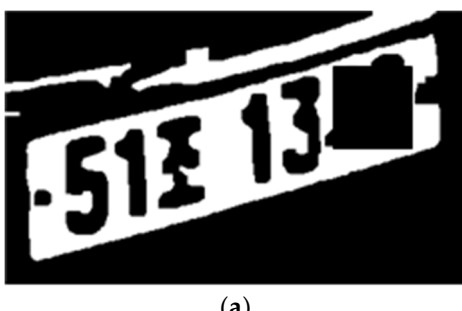
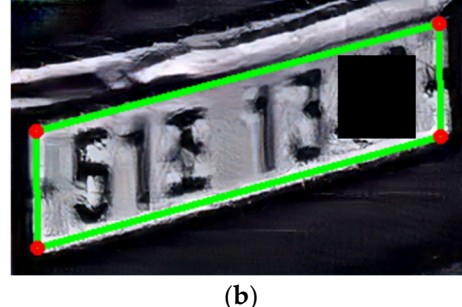

(**a**)　　　　　　　　　　　　　　　　　　　　　　　　　(**b**)

**Figure 7.** (**a**) Input image; (**b**) RANSAC line fitting image.

The next step is to measure the angle of the corners of the license plate-shaped rectangle. If this square corner is not $90°$, it determines that the license plate is distorted, and performs the perspective transformation. Figure 8 shows the image of measuring the angle of the license plate edge.

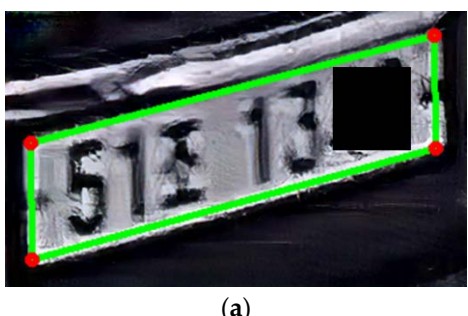
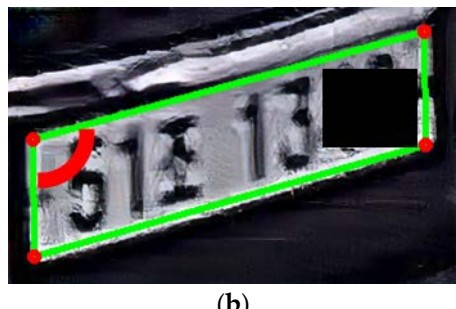

(**a**)　　　　　　　　　　　　　　　　　　　　　　　　　(**b**)

**Figure 8.** (**a**) Input image; (**b**) Angle test image.

### 2.4.4. Perspective Transformation

The shape projected on the image appears differently depending on the angle of the object and camera planes. To map these two planes, the projection transformation is used. If the three points are located on a straight line, the three points are placed on the straight line after the projection transformation.

The projection transformation is a linear transformation expressed as Equation (5).

$$\begin{bmatrix} x' \\ y' \\ 1 \end{bmatrix} = H \begin{bmatrix} x \\ y \\ 1 \end{bmatrix} = \begin{bmatrix} h_1 & h_2 & h_3 \\ h_4 & h_5 & h_6 \\ h_7 & h_8 & h_9 \end{bmatrix} \begin{bmatrix} x \\ y \\ 1 \end{bmatrix} \tag{5}$$

where $H$ is a $3 \times 3$ matrix, $(x, y)$ are coordinates of the image before the transformation and $(x', y')$ are coordinates of the image after conversion. The elements of the matrix $H$ can be calculated using the direct linear transformation (DLT) algorithm [16,17]. To determine the elements of the matrix $H$ using the DLT algorithm, at least five pairs of the corresponding points are required. In this study, the feature points extracted from Figure 9 are used. Thus, the coordinates of the four rectangular vertices were used as feature points on the corresponding projection plane. $H$ was corrected using Equation (6) for distortion correction.

$$x' = \frac{h_1 x + h_2 y + h_3}{h_7 x + h_8 y + h_9}, \; y' = \frac{h_4 x + h_5 y + h_6}{h_7 x + h_8 y + h_9} \tag{6}$$

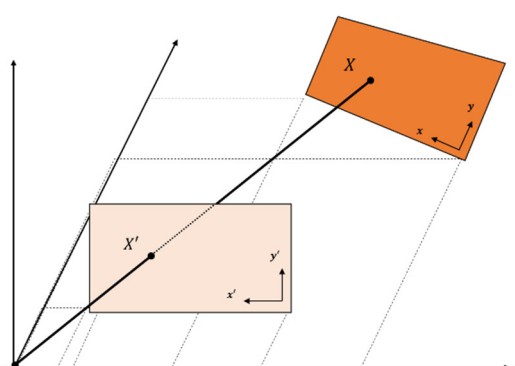

**Figure 9.** Perspective transformation.

Figure 9 shows the result of correcting the perspective distortion using Equation (6).

The input image of Figure 10a shows severe inclination. If this is corrected using a perspective transformation, the position of the license plate becomes constant to favor character recognition (Figure 10b).

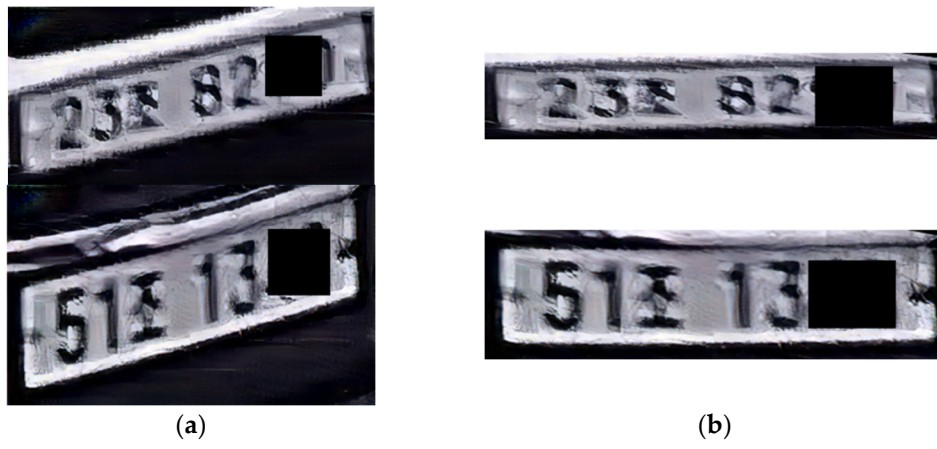

| (a) | (b) |

**Figure 10.** (**a**) Input image; (**b**) Perspective transformation image.

### 2.5. Character Recognition

To confirm the letters and numbers of the image, corrected for perspective distortion, this paper was verified using the YOLO v2 model. This model is commonly used for character and number recognition [18–26]. Figure 11 shows the detection result used for the verification dataset.

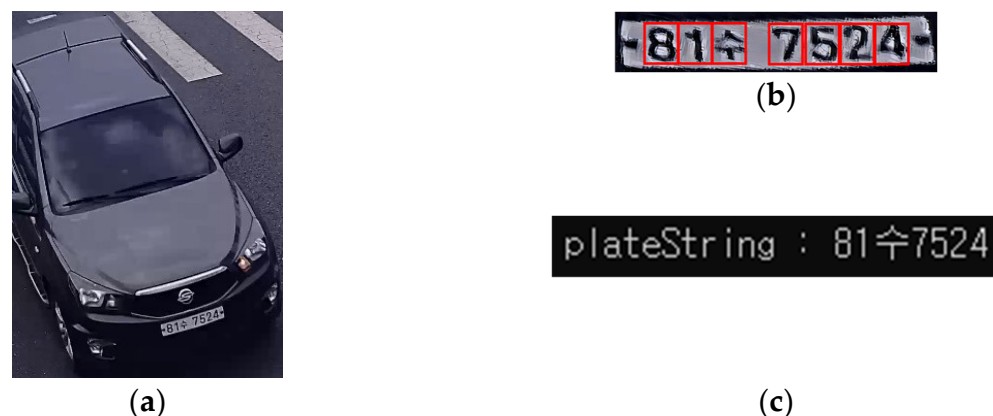

**Figure 11.** (**a**) Input image; (**b**) License plate recognition; (**c**) Recognition result.

## 3. Experimental Results

To evaluate the proposed license plate recognition algorithm performance, we acquired vehicle images using a network camera with a resolution of 3-Mega pixels installed on the road. Still images that can be identified with the naked eye were selected and used as experimental data. The experimental image data consist of 2500 image sheets. The experiment was conducted using license plates with various sizes and colors of characters.

Figure 12 shows the experimental results of using the proposed image improvement algorithm. The experimental results showed that characters misrecognized in existing low-resolution images were successfully recognized using the proposed method. However, there is a problem in that the characters are not normally recognized when the perspective distortion of the license plate is severe among images of the license plate restored in high-resolution. To solve this problem, the distortion-correction technique proposed earlier was applied to images. Figure 13 shows the image to which the distortion-correction technique is applied.

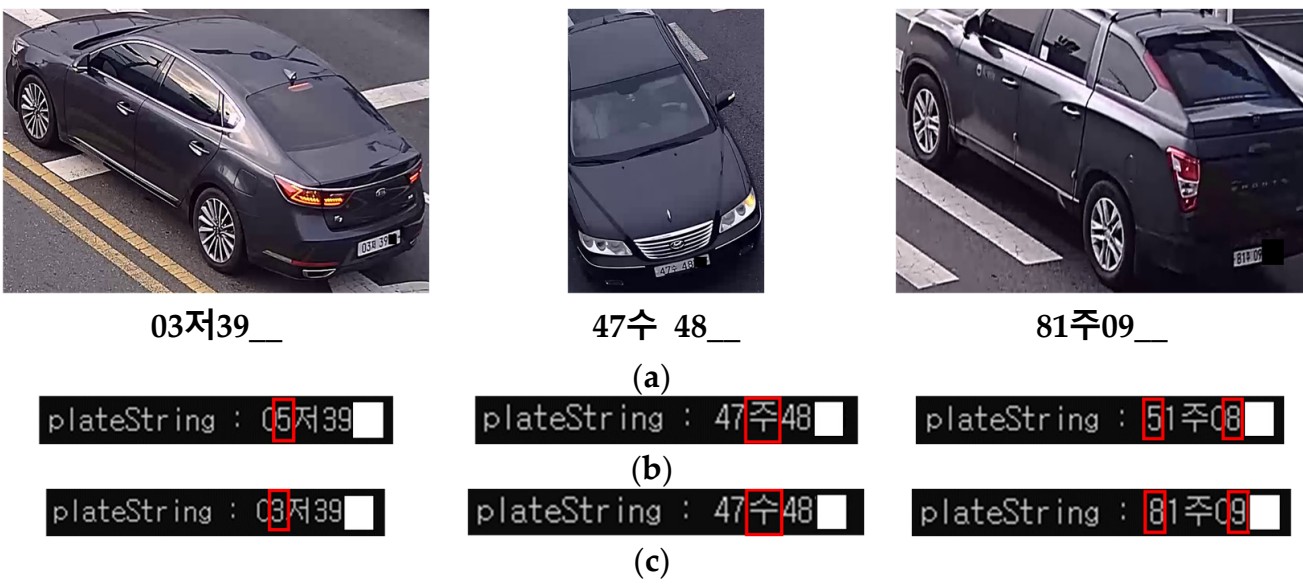

**Figure 12.** (**a**) Input image and Korean string; (**b**) low-resolution result; (**c**) super-resolution result.

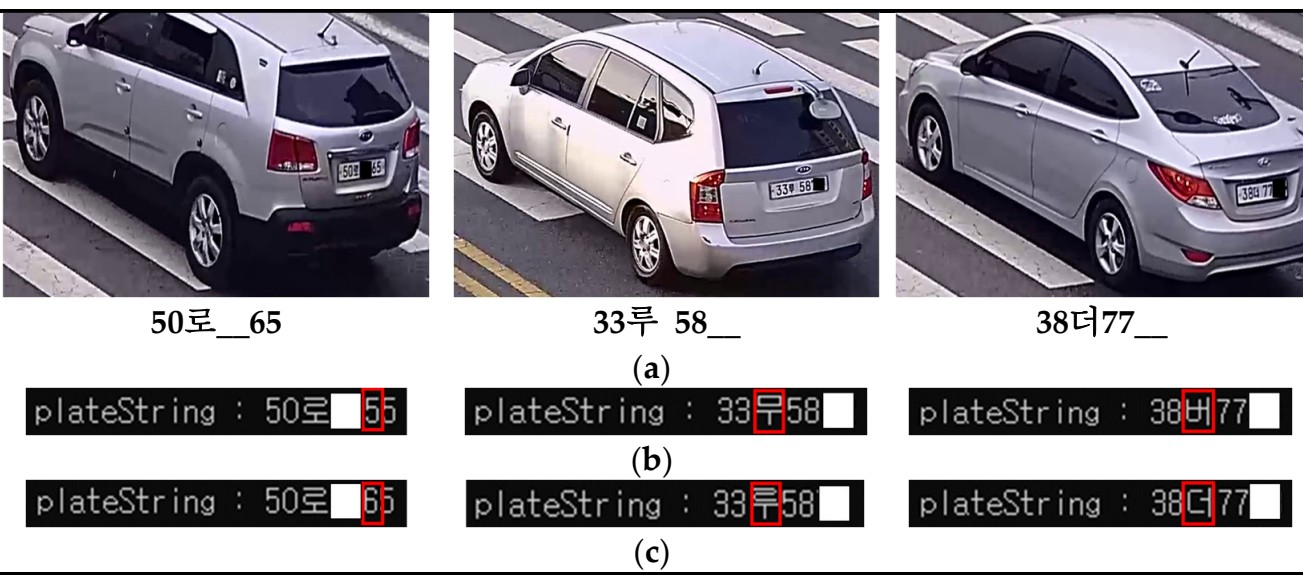

**Figure 13.** (**a**) Input image and Korean string; (**b**) Before perspective transforms; (**c**) After perspective transforms.

Besides the distortion-correction technique, the experimental results verify that the proposed algorithm improves the recognition rate. Table 2 presents the experimental data result using the proposed algorithm. The license plate recognition rate of the original image was 72.6% which was, however, improved to 73.2% by using the low-resolution image reconstruction technique described in Section 2.3. When the distortion correction technique described in Section 2.4 was used in conjunction, the recognition rate became 81.4%, which verified the validity of the proposed license plate detection algorithm.

**Table 2.** Application result of proposed algorithm.

| Total Image | | 2500 |
|---|---|---|
| Original image | True recognition | 1815 |
| | False recognition | 685 |
| | Accuracy | 72.6% |
| Super-resolution method [(2.3)] | True recognition | 1832 |
| | False recognition | 668 |
| | Accuracy | 73.2% |
| Perspective distortion-correction method [(2.4)] | True recognition | 2037 |
| | False recognition | 463 |
| | Accuracy | 81.4% |

### 4. Conclusions

In this study, we proposed a novel method to improve the recognition rate of the deep learning models trained with a general database for vehicle license plate recognition using CCTV images. The proposed method uses the SRGAN method in combination with the perspective distortion-correction algorithm to improve the license plate recognition rate. The SRGAN method was used to improve the low-resolution images acquired by using the CCTV. A perspective distortion-correction algorithm was then deployed for improved recognition of the tilted license plate. The perspective distortion-correction algorithm employs the Otsu threshold method to binarize the image, remove noise from the binarized image, and apply a morphological filter to differentiate license plates. In the acquired image of the vehicle, the license plate area is extracted by using the straight line corresponding to the four sides of the license plate by deploying the RANSAC line fitting technique. The intersection point of the two lines was extracted as a feature point for distortion. The perspective transformation was applied to license plates with severe

perspective distortion using feature points to correct perspective distortion. As a result of performing the character recognition on the corrected image, the character recognition rate in the existing low-resolution image was 72.6%. However, the recognition rate improved to 73.2% using a super-resolution algorithm, and with the SRGAN, the distortion-correction algorithm was also applied for the misrecognized characters and the recognition rate was further improved to 81.4%. The validity of the proposed algorithm was verified by confirming a higher-recognition rate improvement of more than 8.8% compared to low-resolution images. We anticipate that in our future work, we will also associate the smart parking area implementation. Further study should consider the combination of lighter deep neural networks to maintain the accuracy and improve recognition speed.

**Author Contributions:** T.-G.K., B.-J.Y., Y.J. and H.D.K. conceived and designed the experiments; T.-G.K. and T.-H.K. performed the experiments; T.-G.K., B.-J.Y., Y.J. and H.D.K. analyzed the data; T.-G.K. and J.-Y.L. contributed analysis tools; T.-G.K. and K.-H.P. wrote the paper. All authors have read and agreed to the published version of the manuscript.

**Funding:** This work was supported by the Technology development Program(S2897132) funded by the Ministry of SMEs and Startups (MSS, Korea). This research was supported by Basic Science Re-search Program through the National Research Foundation of Korea (NRF) funded by the Ministry of Education (NRF-2018R1D1A1B07040457).

**Institutional Review Board Statement:** Not applicable.

**Informed Consent Statement:** Not applicable.

**Data Availability Statement:** Not applicable.

**Acknowledgments:** This work was supported by the Technology development Program(S2897132) funded by the Ministry of SMEs and Startups (MSS, Korea). This research was supported by Basic Science Research Program through the National Research Foundation of Korea (NRF) funded by the Ministry of Education (NRF-2018R1D1A1B07040457).

**Conflicts of Interest:** The authors declare no conflict of interest.

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
