# Peer review of "Recognition of Vehicle License Plates Based on Image Processing"

_applsci, doi:10.3390/app11146292_

Round 1

Reviewer 1 Report

Thank you for your efforts to address all my comments/suggestions.

I found the responses quite satisfactory, and the revised version has been much improved.

Author Response

Thank you for your opinion. 

It seems that the readability and quality of the manuscript are improving.

Reviewer 2 Report

Thanks for the revision, however, there are still a few issues that need to address. The first concern is the quality of the literature review, which is not appropriate for a journal paper. Authors should review and discuss and possibly highlight previous studies' shortcomings and highlight if their proposed method can address this matter.  The novelty/contribution of the paper needs to be highlighted within the following section as well. 

The text within Figure 3 is hard to read. 

Also, could authors add a high-level flowchart of their method to show how images are going to be processed? 

Author Response

(The authors gave the same response as above.)

Reviewer 3 Report

In this study, we proposed an algorithm to solve the problem that occurs when recognizing a vehicle license plate acquired by closed-circuit television (CCTV) using a deep learning model trained with a general database. The deep learning model provided has the disadvantage that it recognition rate is low in low resolution and tilted images because it is trained with images acquired in the front side of the license plate. Vehicle images acquired by CCTV have problems, such as limitation of resolution and perspective distortion. Thus, it is not easy to apply a provided deep learning model. To solve the low-resolution problem, a super-resolution generative adversarial network (SRGAN) was deployed. The authors should address the following comments to improve the presentation of their manuscript.

English shod be revised there are a lot of typo mistakes.

  • To solve the low-resolution problem, a super-resolution generative adversarial network (SRGAN) did not mention how he develops model fusion?? cascade learning??
  • Authors shod add accuracy in number at end of the abstract.
  • Experimental Setup??  Authors should provide to the reader all the parameters that have been used to generate the numerical results. At Present, a few of the parameters are missing.
  • Authors need to show the GitHub link or implementation details that should be mentioned in the paper that will be good readers.
  • the conclusion part is missing with future direction. Any possible downside of the proposed approach??.
  • Introduction and related work should be separate which will show the importance of this study and good for the readers.AI>ML>DL>Proposed approach.

da Rocha Gesualdi, A., & de Seixas, J. M. (2002, November). Character recognition in car license plates based on principal components and neural processing. In VII Brazilian Symposium on Neural Networks, 2002. SBRN 2002. Proceedings. (pp. 206-211). IEEE. Deeba, F., Zhou, Y., Dharejo, F. A., Khan, M. A., Das, B., Wang, X., & Du, Y. (2021). A plexus‐convolutional neural network framework for fast remote sensing image super‐resolution in wavelet domain. IET Image Processing. Ledig, C., Theis, L., Huszár, F., Caballero, J., Cunningham, A., Acosta, A., ... & Shi, W. (2017). Photo-realistic single image super-resolution using a generative adversarial network. In Proceedings of the IEEE conference on computer vision and pattern recognition (pp. 4681-4690). Irani, Michal, and Shmuel Peleg. "Super resolution from image sequences." [1990] Proceedings. 10th International Conference on Pattern Recognition. Vol. 2. IEEE, 1990.

  • Table 1 Application result of proposed algorithm should include the Reference no.
  • If the data volume is very large, the transformation will become a super-slow process. Please give out the details about the transformation and address how to handle a large amount of data in, a super-resolution generative adversarial network (SRGAN). It looks confusing for readers
  • English shod be improved there are grammatical errors.
  • The Paper has a lot of formatting issues.

.

Round 2

Reviewer 1 Report

Thank you for your efforts to address all my comments/suggestions.

I found the responses quite satisfactory, and the revised version has been much improved.

Reviewer 3 Report

The authors did excellent work this paper looks very interesting for the readers so I accept this paper for publication in the present form.

Author Response

Thank you for your opinion.

It seems that the readability and quality of the manuscript are improving.

This manuscript is a resubmission of an earlier submission. The following is a list of the peer review reports and author responses from that submission.

Round 1

Reviewer 1 Report

The manuscript presents an interesting topic, and it is easy to read. Authors present a slightly new approach for automatic license plate recognition.

There are no big issues with English language.

Authors should improve the state of art and the bibliography since there are a lot of similar work in the literature.

In the Introduction you say “A public database-based learning model provided to recognize vehicle license plates based on deep learning is a model learning based on the image of the vehicle license plate from the front.”. This is true considering the dataset used on reference 3. However, from reference 2 you could obtain a dataset from CCTV images. Also, there are several datasets from CCTV cameras available from other countries that could be used to test your approach.

Construct a new database with images similar to those acquired from CCTV is not so hard and would be a great contribution.

In line 56 you say “The captured image using CCTV is low-resolution and tilted.”. Currently CCTV systems have high resolution cameras. I suppose you meant to say that, in CCTV images, the license plates have low resolution, and they are tilted.

The kernel size used in the morphological operations is fixed: 3x3 for dilation and 6x6 for erosion. How you get these values? Trial and error? I have some concerns regarding their effectiveness for different ROI sizes.

Authors must not start a section or subsection with a figure (e.g. subsections 2.1 and 2.3.1).

Figures should appear after their citation on text (e.g. Figure 1, Figure 3).

Reviewer 2 Report

General comments:

This article laid out a workflow to detect license plates based on CC-TV data. While the topic is interesting, there are serious flaws associated with the current version of this document. Firstly, previous studies are not reviewed to highlight and find the current contribution of the article. Besides, the methodology is not clearly described, and it is not clear why authors have used GANs and what its benefit in comparison to more traditional methods. Besides these general comments that need to be addressed, please see my detailed comments below.  

Detailed comments:

Line 2: Perhaps update the title to “Recognition of Vehicle License Plates based on Image Processing”.

Line 15: Deep learning model?

Lines 15 to 17: If a deep learning model developed based on the correct set of images, it likely can detect license plates similar to that of nondeep learning-based methods. Therefore, this statement is generally, incorrect. Please update this statement.

Lines 17 to 19: Similar to the previous comment, what if a preprocessing step fixes the distortion of the images?

Lines 21 to 23: Please rewrite the scope and contribution of the paper as in its current format, it is hard to identify the paper contribution.

Line 30: CC-TV transmit images or video? It is correct that videos are essentially made of multiple images per second, but the sentence suggests that CC-TV cameras are taking only images, which is not accurate.

Line 31: avoid using buzz words like big data.

Line 33: There are many automated license plate recognition systems that police used across the world. I would recommend authors rewrite this sentence.

Lines 46 to 48: The sentence is hard to read and understand, please rewrite the sentences here.

Lines 48 to 49: A reference is needed to back up this claim.

Lines 56 to 61: Please clarify if the paper is using previously developed architectures to create a new workflow.

Line 68ff: Figure should come after a description, please place Figure 1 accordingly. Also, each step of workflow needs to be references to a particular section of the article that describes the details of the process. The workflow is also not complete as only processes are shown. The flowchart should show what sort of data are inputted what is the outcome (at least at a high level).

Lines 91: Can authors clarify if they have developed the SRGAN themselves? If not, this needs to be referenced. Also, I cannot see any specific literature review section.

Line 110: A simple image sharpening method can also improve the images.

Line 224ff: Please report the performance measures consistently. Currently, some values are reported as percent value while others are different. Also, is 76 percent acceptable in comparison to other studies?

Reviewer 3 Report

My main concern is that the proposed methodology is built without well explained reasons. It appears as a collage of pieces taken here and there, but it is not clear why precisely those. For examples, why the YOLO v2 model has been used? Are there any benefit using it? How does the performance of the proposed method change using most appropriate classification model? Similarly, they propose  to use SRGAN network architecture, without explaing why it instead of others. 

Moreover the Introduction is very poor, there is an extensive literature on the subject that authors seem to ignore, as also highlighted by the poor list of references.